# Pauli Exclusion Classical Potential for Intermediate-Energy Heavy-Ion Collisions

**Claudio O. Dorso** [1,†] , **Guillermo Frank** [2,†] **and Jorge A. López** [3,*,†]

1   Departamento de Física, FCEN, Universidad de Buenos Aires, Buenos Aires 1428, Argentina
2   UIDI, Universidad Tecnológica Nacional, Buenos Aires 1407, Argentina
3   Department of Physics, University of Texas at El Paso, El Paso, TX 79968-0515, USA
*   Correspondence: jorgelopez@utep.edu; Tel.: +1-915-747-7528
†   These authors contributed equally to this work.

**Abstract:** This article presents a classical potential used to describe nucleon–nucleon interactions at intermediate energies. The potential depends on the relative momentum of the colliding nucleons and can be used to describe interactions at low momentum transfer mimicking the Pauli exclusion principle. We use the potential with molecular dynamics to study finite nuclei, their binding energy, radii, symmetry energy, and a case study of collisions.

**Keywords:** nucleon–nucleon interactions; intermediate-energy heavy-ion collisions; nuclear symmetry energy

## 1. Introduction

In recent years, the field of nuclear physics has witnessed an explosion in interest on a large variety of nuclear phenomena, ranging from the structure of nuclei and their decay modes, passing through the study of exotic nuclei, and up to the properties of neutron stars. In particular, the study of heavy-ion collisions at intermediate energies requires the use of theoretical models, which are the main tool to extract information from such reactions.

Most models intend to reproduce reactions. Until now, it has been a trade-off; while most researchers study nuclear reactions using models in which the nucleons (or their *avatars*) interact via an average mean field, a few others use classical dynamics with direct nucleon–nucleon interactions. Those using mean field methods sacrifice clusterization, fragment formation, and many other critical phenomena for the inclusion of some quantum and semi-quantum features. On the other hand, those using classical dynamics fail in including quantum features but gain in preserving nucleon–nucleon correlations, statistical fluctuations, phase changes, and all critical phenomena of the upmost importance in the later stages of fragmentation reactions.

In an odd complementarity, the quantum aspects of mean field models aid in achieving a proper energy distribution through the early stages of the reaction, while failing at fragmenting the system in the latter stage of the collision, whereas the classical methods excel in the latter phase but cannot guarantee proper reaction dynamics in the early part of the reaction.

By themselves, different mean field models from the two basic families, Boltzmann–Uhling–Uhlenbeck (BUU) and Quantum Molecular Dynamics (QMD), show discrepancies in their predictions. The results of a recent comparison of different models indicate that they vary in the stability of the initialized nuclei, the effectiveness of Pauli blocking in nucleon–nucleon collisions, as well as other predicted flow observables, mostly due to differences in the initialization of the systems, the treatment of the collision integral, and other much-entangled effects [1].

The latter are probably due to the operator's selections of number of "test particles" (i.e., the avatars) and, more importantly, to the artificial fluctuations included by hand both

in QMD and in the Langevin framework of BUU. Although fragmentation was not studied in such comparison, it continues to be the Achilles' heel of all such models [2–6]; to compare to final yields, final matter density distributions must be used as input to "afterburner" codes, such as the Statistical Multifragmentation Model [7], and the like; for a description of the Antisymmetrized Molecular Dynamics (AMD) model with an added mechanism for fragment formation, see [8], and, for a recent application of AMD and SMS, see [9].

On the other hand, in spite of violating quantum principles, classical dynamics models (such as classical molecular dynamics) are capable of reproducing both the out-of-equilibrium and the equilibrium parts of a collision. Indeed, classical molecular dynamics (CMD) models are able to describe non-equilibrium dynamics, hydrodynamic flow, and changes of phase without adjustable parameters, neck fragmentation [10], phase transitions [11], critical phenomena [12,13], caloric curve [14,15], and isoscaling [16] in nuclear reactions, as well as in the formation of nuclear pasta in infinite systems [17–19].

[A particular success of CMD over mean field models is the calculation of symmetry energy at clustering densities and temperatures [20], which showed good agreement with experimental data [21–23]. This corroborated the Natowitz conjecture that the asymptotic limit of $E_{sym}$ would not tend to zero at small densities as predicted by mean-field theories.]

In spite of such successes, CMD lacks all quantum effects which can affect the reaction dynamics in, at least, two fronts: its energy distribution and wave mechanics. Nucleons in bound nuclei have discrete energy levels, as ruled by Fermi–Dirac statistics, with their occupation regulated by the Pauli exclusion principle. A second effect is the lack of wave features of the particles, which becomes dominant whenever the mean interparticle distance is smaller than the mean thermal de Broglie wavelength. Although the limits of such deficiencies have been estimated [24], and tend to disappear at sub-saturation density and sub-critical temperatures, their effects in the various stages of the collision, especially those that happen at saturation and higher densities and cold temperatures, are not known. A formulation to correct for these inadequacies is the topic of this work.

In this work, we first introduce, in Section 2, a classical two-boy potential that mimics the effect of the Pauli exclusion principle by de-enhancing interactions between nucleons that are too close in phase space. We then use such potential, in Section 3, to construct "nuclei" with binding energies and radii close to the experimental values, and, in Section 4, the symmetry energy of such nuclei is studied. Next, in Section 5, we test the usefulness of the constructed nuclei to simulate nuclear reactions. We finally present a series of observations in Section 6, along with an outlook for future uses of the potentials developed.

## 2. Nuclear Potentials and the Quantum Problem

Classical dynamical methods use point particles to represent nucleons interacting through pair potentials. This approach started decades ago and has advanced ever since [17,25–32]. Classical dynamics has evolved from using a common mean field plus a residual scattering in an adaptation of Nordheim's propagation of individual nucleons [33], to a full fledge incorporation of nucleon–nucleon potentials [26,27,32,34] created ad hoc to mimic nuclear properties, such as binding energy, saturation density, and nucleon–nucleon scattering cross section, among others. Some of the potentials used to study nuclear reactions can be found in [24,25,27,28,34–40], while those that have been used to study infinite nuclear systems (as expected to exist in neutron stars) are [17,29–31,35–38].

Reiterating, in bound clusters, such as cold nuclei, individual nucleons attain discrete energies distributed according to Fermi–Dirac statistics, with the occupation of such levels regulated by the Pauli exclusion principle. Although at high excitation energies the number of levels available for the nucleons increases exponentially and render Pauli blocking practically obsolete [24], at lower temperatures, the occupation of levels increases and the nucleon dynamics faces prohibiting energy and momentum transfers in nucleon–nucleon collisions. Since in heavy-ion collisions at intermediate energies the initially cold nuclei will compress to supra-saturation densities, Pauli blocking is expected to play a role in suppressing some of the nucleon–nucleon interactions. As the compressed system heats up

to a few MeVs, and the energy level density becomes more dense, the system will approach a classical level. Later, when the system cools and expands, Pauli exclusion will again regulate final-stage interactions, including fragmentation.

This suggests that, for a classical potential to mimic the Pauli blocking, it should try to reduce the number of collisions occurring in cold nuclear systems, i.e., at low momentum transfer, or in near proximity in phase space.

### 2.1. Adding Pauli Blocking to a Potential

One way to simulate Pauli blocking in classical collisions is to introduce a momentum-dependent repulsion that would de-enhance nucleon–nucleon collisions at low energies and low momentum transfer. This approach was first taken by Wilets et al [26], followed up by Dorso and Randrup [41], and, in other approximation, by Boal et al. [42,43]. Subsequent efforts of Dorso and Randrup [28] managed to simulate the Pauli exclusion principle, provided nucleons with a realistic ground-state Fermi motion, but failed at endowing the model with an appropriate repulsion between equal nucleons.

Recently, in a study preliminary to this one [44], we followed the footsteps of Dorso and Randrup, and introduced a momentum-dependent repulsive potential that gradually reduces the strength of the nuclear potential as a function of the relative momentum of the interacting nucleons. The nuclear part of the potential was crafted with Lennard–Jones-like two-body interactions plus a standard Coulomb repulsion between protons. The parameters of the potential were adjusted, once and for all, to yield infinite systems with the proper saturation energy and compressibility over a broad range of temperatures and densities. The resulting potential managed to solve the problem of [28], namely, avoiding the formation of di-neutron structures, but, in spite of such improvements, when compared to experimental values, the resulting radii of bound nuclei of [44] were underpredicted by a large 15%, and the binding energies were overbound by a huge 50%.

Concluding, the motivation for this study is to develop a model that would maintain the advantages of classical models, respect the Pauli exclusion principle, and prevent the formation of nonphysical states, while yielding stable nuclei with the correct binding energy and radii. In this work, we present a potential that solves these problems, and use it to calculate the radii, binding energy, symmetry energy, and demonstrate its usefulness in studying reactions, their evolution, and the calculation of final multiplicities.

### 2.2. The Potential

The proposed potential consists of a nuclear part, a "Pauli" part, and a Coulomb part. The Pauli potential is inspired on the work of Dorso and Randrup [41], and the nuclear part on [28]. The Pauli depends on the relative momenta of the interacting nuclei. Defining a dimensionless distance in phase space between any pair of nucleons $i$ and $j$ as $s_{ij}^2 = p_{ij}^2/p_0^2 + q_{ij}^2/q_0^2$, where $q_{ij}^2 = |\vec{r}_i - \vec{r}_j|^2$, and $p_{ij}^2 = |\vec{p}_i - \vec{p}_j|^2$, and the parameters $p_0$ and $q_0$ determine the volume in phase space that is excluded around each particle; the values of $p_0$ and $q_0$ are adjusted as to satisfy the uncertainty relation, $p_0 q_0 \approx 2\hbar$. The Pauli part of the potential is:

$$V_{Pauli}(q_{ij}, p_{ij}) = V_0 e^{-\frac{1}{2}\left(q_{ij}^2/q_0^2 + p_{ij}^2/p_0^2\right)} \tag{1}$$

with the scale factors $V_0 = 5.165$ MeV, $p_0 = 61.969$ MeV/c, and $q_0 = 6.0$ fm.

The strength of the $V_0$ was selected to attain the proper Fermi gas values. Notice that, different from our previous work [44], this potential does not have a cut-off distance to avoid discontinuities of long-range interactions occurring in collisions. It is worth mentioning that, although the value $q_0 = 6$ fm may give the impression that $V_{Pauli}$ is nearly a constant within the nucleus and does not have much of an effect, this is not the case. As it will be seen in Section 3, the nucleus diameter (c.f. Figure 1) is substantially larger than $q_0$, and, furthermore, in collisions, the scale for nucleon–nucleon interactions is set by the entire volume of the collision. In addition, it is important that $p_0 q_0 = 1.88\hbar$, very close to the uncertainty relation, and slightly smaller than $2\hbar$ as expected from geometric

considerations [41]. Other authors have used other values of $q_0$ (see, e.g., [45]), but this give-and-take game yields values of $p_0$ that lie outside the range of interest for building static nuclei and for simulating collisions; we, thus, did not find strong reasons for varying the parameters used in the originals works [28,41].

In summary, the potential adds a modulating term that introduces a repulsive force between equal nucleons at short values of $p_{ij}$ and $q_{ij}$, thus forbidding their interaction. Indeed, at $r = 0$ and $p = 0$, the potential (1) becomes $V_{Pauli} \rightarrow V_0$, generating a kinetic energy resembling the "Fermi energy" expected under those circumstances. The blue line of Figure 2 shows the Pauli potential for the case of zero momentum transfer, $p_{ij} = 0$.

The nuclear part of the potential is:

$$V_{Nuclear}(q_{ij}) = \begin{cases} V_{np}\left[\left(\frac{\sigma}{q_{ij}}\right)^n - \left(\frac{\sigma}{q_{ij}}\right)^m\right] & \text{neutron - proton} \\ V_{pp}\left(\frac{\sigma}{q_{ij}}\right)^n & \text{proton - proton} \\ V_{nn}\left(\frac{\sigma}{q_{ij}}\right)^n & \text{neutron - neutron} \end{cases} \quad (2)$$

where the parameters $V_{np} = 18.0$ MeV and $V_{pp} = V_{nn} = 18.0$ MeV were determined by fitting the binding energies and radii of atomic nuclei. The other parameters, $\sigma$, $m$, and $n$, were determined from the experimental value of the nuclear binding energy of the $^4He$ nucleus (as explained in [44]) to be $\sigma = 1.75$ fm, $n = 8$, and $m = 3.4$, which correspond to a binding energy at saturation density of $E(\rho_0) = -16$ MeV. The fitting of the parameters $V_{np}$, $V_{pp}$, and $V_{nn}$ was obtained by a random walk in the parameter space, starting from the values of [46], while nuclei of various sizes were crafted (with the procedure explained in Section 3), and their binding energy and radii were calculated, until a good agreement with experimental data of [47] was obtained.

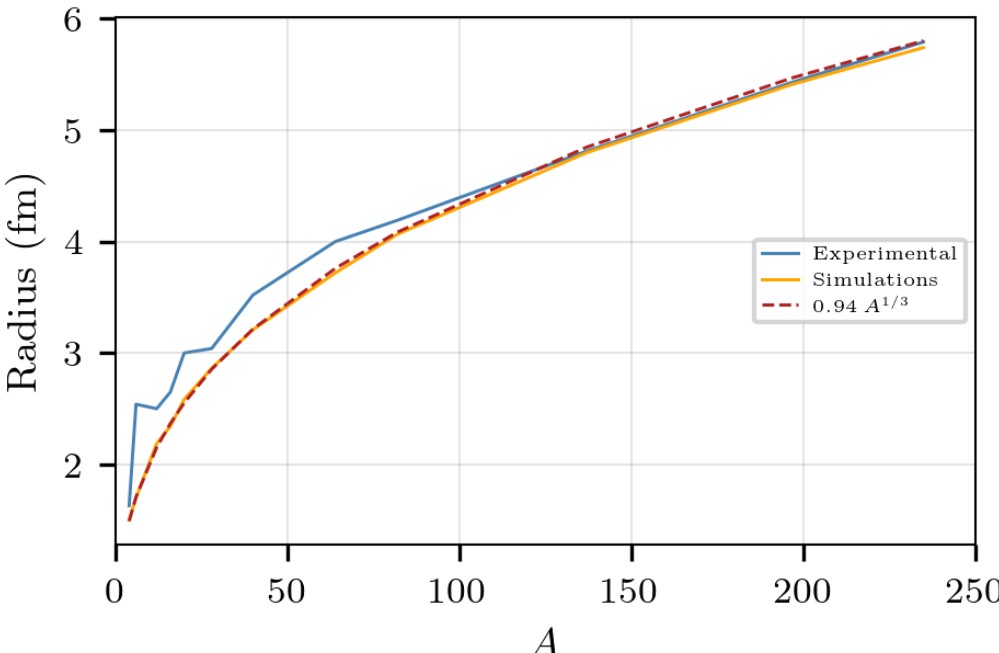

**Figure 1.** Radii of simulated nuclei compared to experimental data. The data points in blue correspond to commonly accepted experimental values [47].

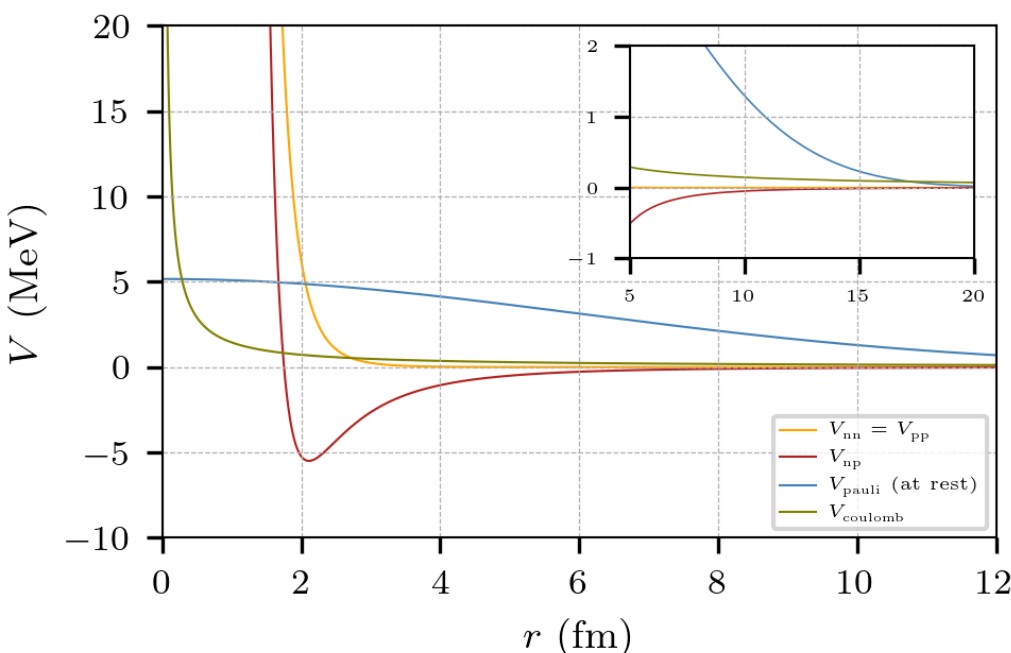

**Figure 2.** Nuclear, Pauli, and Coulomb potentials. The blue line corresponds to the spatial factor of the Pauli potential, $V_{Pauli}(r, p = 0)$, plotted as a function of $r$.

It is worth mentioning that the neutron–neutron term of the potential (2) was introduced to prevent the formation of di-neutrons, while the proton–proton one was incorporated only to preserve isotopic symmetry, as the formation of di-protons is impeded by $V_{Coulomb}$.

Finally, for the Coulomb potential, we use the point-charge repulsion:

$$V_{Coulomb}(r) = \frac{e^2}{r} .$$ (3)

where $e$ is the elementary electric charge, and thus $e^2 = 1.44$ MeV fm. These potentials are shown in Figure 2.

### 3. Nuclei

The previous potential was used to create clusters of nucleons to simulate nuclei. This was achieved by confining nucleons inside a three-dimensional quadratic external potential, and allowing them to evolve by molecular dynamics. Initially, the nucleons were endowed with momenta corresponding to a temperature $T \approx 1.0$ MeV according to a Maxwell distribution, and were cooled down gradually by reducing their momenta until reaching a temperature of $T \approx 0.01$ MeV, at which point the external potential was removed. For the fitting of the nuclear parameters, the heating-cooling procedure was repeated once to make sure that the nuclei were self-stable, and to obtain robust averages of the binding energy and radii.

Different from our previous study [44], here we use classical molecular dynamics with potentials (1)–(3), instead of the Metropolis–Monte Carlo (MMC) method [48]; the MMC method, however, was used only at the end to verify that both methods yielded the same final binding energy of the cold nuclei. It must be observed that the Pauli potential (1) is not separable, and the usual symplectic integrators of the equations of motion used in molecular dynamics cannot guarantee the conservation of energy. Fortunately, for the present case of molecular dynamics with thermostats, the energy is obviously not conserved. For the calculation of reactions, as presented in Section 5, the integration occurs over short times, which has been shown to limit the divergence of conserved quantities [49].

The radii of several nuclear-like clusters bound with the potentials (1)–(3) are presented in Figure 1 and compared to experimental values. The radii were calculated as the r.m.s value of the position of the nucleons in the "ground state" ($T \approx 0.01$ MeV) of the nuclei constructed with the procedure presented before. The figure also includes the curve $R = c\,A^{1/3}$ that corresponds to the best fit of the radii of the simulated nuclei.

Table 1 and Figure 3 show the binding energy per nucleon corresponding to the nuclei of Figure 1. The results of the simulation show a much better agreement with the experimental data of [47], and a large improvement with respect to the previous model [44]. For comparison with other classical models, see the Simple Semi-classical Potential of Horowitz and coworkers [30], and that of Dorso and Alcain [50].

**Table 1.** Binding energy per nucleon and radius of selected nuclei.

| Element | A | N | Z | E/A [MeV] | R [fm] |
|---|---|---|---|---|---|
| Helium | 4 | 2 | 2 | 5.23 | 1.5 |
| Lithium | 6 | 3 | 3 | 5.58 | 1.7 |
| Carbon | 12 | 6 | 6 | 6.64 | 2.19 |
| Oxygen | 16 | 6 | 10 | 6.97 | 2.34 |
| Neon | 20 | 10 | 10 | 7.38 | 2.58 |
| Silicon | 28 | 14 | 14 | 7.78 | 2.86 |
| Calcium | 40 | 20 | 20 | 8.10 | 3.21 |
| Zinc | 64 | 34 | 30 | 8.76 | 3.72 |
| Krypton | 82 | 46 | 36 | 8.51 | 4.06 |
| Cesium | 137 | 82 | 55 | 8.30 | 4.80 |
| Mercury | 197 | 117 | 80 | 7.98 | 5.41 |
| Uranium | 235 | 143 | 92 | 7.65 | 5.74 |

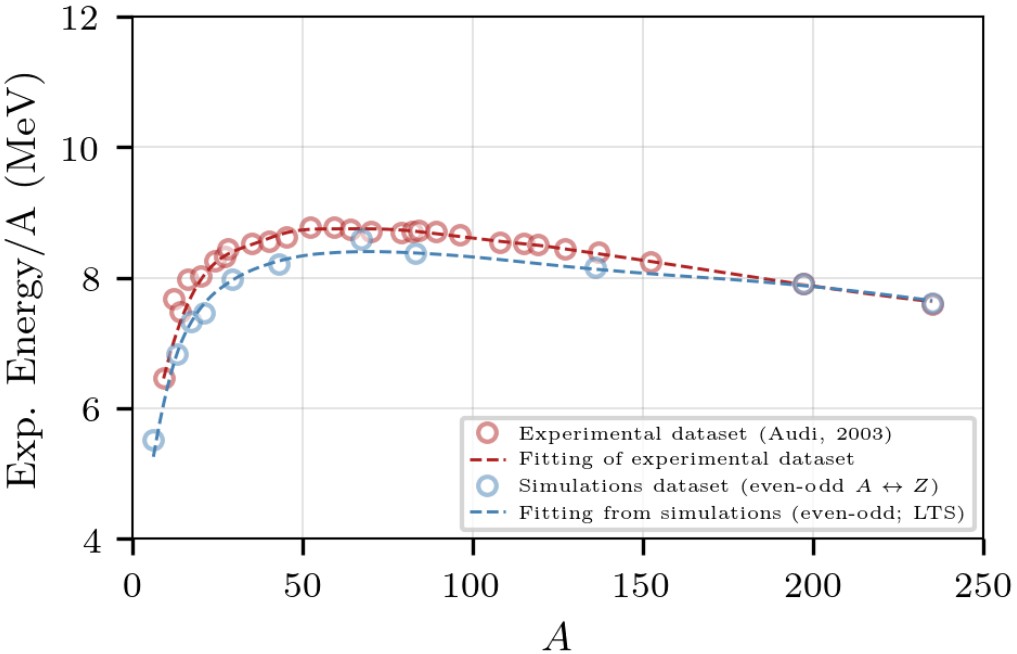

**Figure 3.** Binding energy obtained for the nuclei of Figure 1. The blue dashed line is the fit to the energies of the simulated nuclei using the least-trimmed-square estimator (LTS) [47].

The figure also shows a semi empirical mass formula fit of the values of the form

$$E/A = C_v - C_s A^{-1/3} - C_c Z^2 A^{-4/3} - C_{sym}(A - 2Z)^2 A^{-2}\,, \qquad (4)$$

with the coefficients shown in Table 2; the values were obtained by fitting the semi empirical mass formula to the binding energies of the simulated nuclei using the least-trimmed-square

estimator (LTS). The resulting fit corresponds to the blue dashed line of Figure 3. The values of Audi [47], Rholf [51], Dorso [39], and of the Generalized liquid drop model [52] are presented for comparison.

**Table 2.** Coefficients of the semi empirical mass formula.

| Coefficient | This Work | Audi [47] | Rholf [51] | Dorso [39] | Generalized Liquid Drop Model [52] |
|---|---|---|---|---|---|
| $C_v$ | 13.98 | 15.550 | 15.75 | 17.37 | 15.81 |
| $C_s$ | 15.12 | 17.109 | 17.8 | 14.38 | 18.54 |
| $C_c$ | 0.49 | 0.711 | 0.711 | 0.226 | 0.714 |
| $C_{sym}$ | 21.46 | 21.110 | 23.7 | 25.08 | 23.599 |

It must be mentioned that the Coulomb and symmetry terms of Equation (4) depend explicitly on A and Z, which introduces discontinuities for different isotopes of the same element, and such fluctuations are more noticeable around $A \approx 70$. Because of this double A–Z dependence, the fit required a particular procedure which is summarized in Appendix A.

## 4. Symmetry Energy

The variation of the binding energy as a function of the isotopic number became relevant when radioactive beam facilities became able to produce nuclei away from the stability valley. Such variation, known as the nuclear symmetry energy, is needed to study topics ranging from nuclear structure to astrophysical processes [53].

Historically, Weizsäcker introduced an asymmetry term to his 1935 parametrization of the nuclear binding energy to enhance binding of nuclei with an equal number of protons and neutrons [54]. When the mass formula was generalized to be density-dependent, such term was modified to include the role of isospin in the density-dependent asymmetry term, $E_{sym}(T, \rho)$ [55]. The symmetry energy is defined as:

$$E_{Sym}(\rho, T) = \frac{1}{2!} \left[ \partial^2 E(\rho, T, \alpha) / \partial \alpha^2 \right]_{\alpha=0} , \tag{5}$$

with $\alpha = (N - Z)/(N + Z) = 1 - 2x$ and with $x = Z/(N + Z)$.

In this case, the symmetry energy can be evaluated from the binding energies of the isotopic nuclei data constructed by the method presented before. For this purpose, several instances of a specific nucleus are constructed, and the average binding energy is obtained through $\langle E/A \rangle = \frac{1}{N} \sum_N E/A$, where $N$ is the number of nuclei constructed. Figure 4 illustrates the convergence of $\langle E/A \rangle$ as $N$ increases, for three different isotopes of $Cs$. In what follows, $N$ is set to 20.

Figure 5 shows the average values of the binding energies of the isotopes $^{127}Cs$, $^{129}Cs$, $^{131}Cs$ and $^{133}Cs$, as a function of the proton asymmetry $x$; the values were obtained as averages of 20 constructions of the isotopes. For comparison, we also present the experimental values for isotopes of $Cs$ taken from [56]. In addition, a quadratic fit obtained by least squares is shown, $E/A = 353.2x^2 - 301.8x + 56.18$, from which the symmetry energy can be obtained to be $E_{sym} = 353.2/4 = 88.3$ MeV. This value is not too distant from the symmetry energy of 70 MeV obtained in a previous molecular dynamics study of nuclear matter at $T \approx 0$ MeV and density $\rho \approx \rho_0/2$ [18], but it is outside the generally accepted values of 30–50 MeV obtained with microscopic field theories for isospin symmetric matter at saturation density [53,57]. In passing, we use the same method to fit the experimental data and extract the corresponding symmetry energy, the resulting fit is $E/A = 90.48x^2 - 77.74x + 8.28$, and its symmetry energy is $90.48/4 = 22.62$ MeV, somewhat below the accepted range from microscopic field theories.

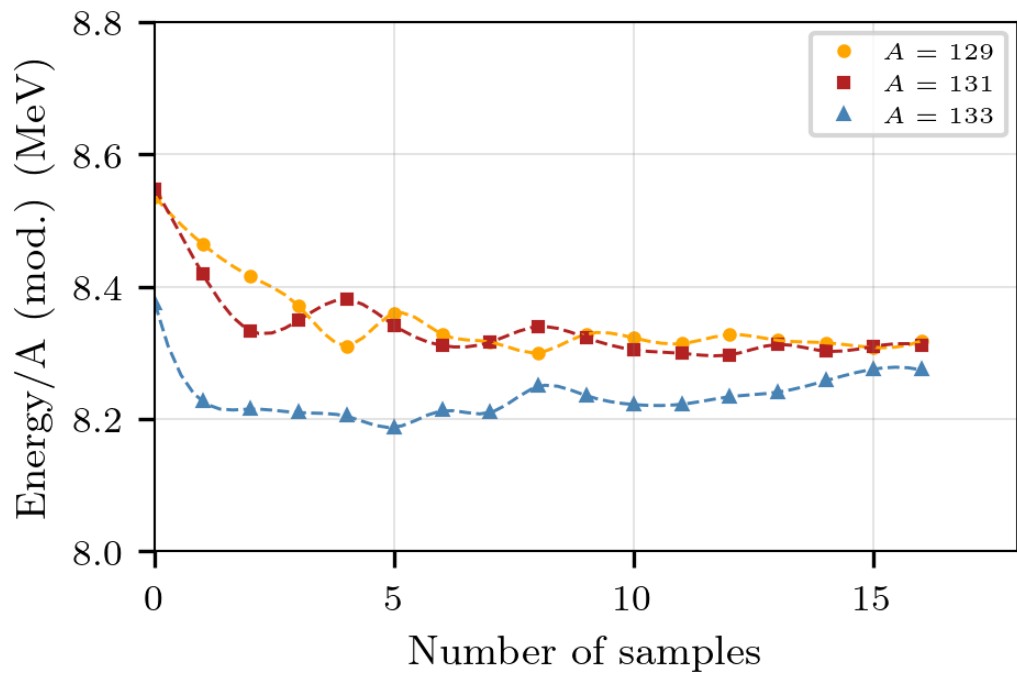

**Figure 4.** Convergence of the magnitude of the average binding energy of various isotopes of *Cs* as a function of the number of samples.

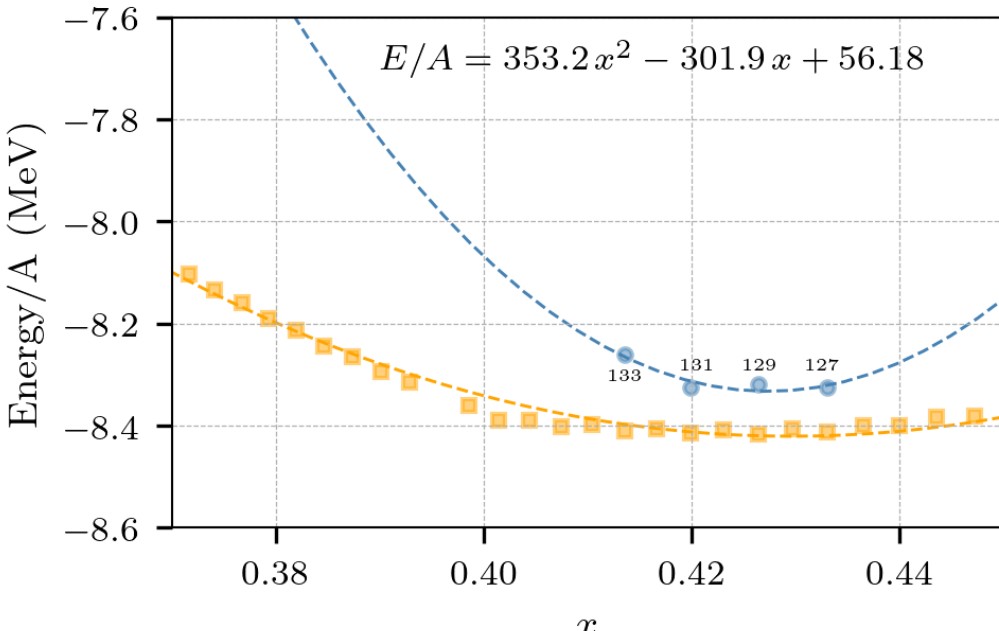

**Figure 5.** Average values of the binding energy of various isotopes of *Cs* obtained from simulations (blue) and least-squares fit. In addition, the experimental values of the binding energies of *Cs* isotopes (orange) and a corresponding quadratic fit are shown.

## 5. Collisions

The ultimate goal of any model is to reproduce experimental observables, which, in the case of nuclear physics, are obtained through reactions. In this last section, we present a first attempt to reproduce nuclear collisions with "nuclei" created as described in Section 3 with the potentials shown in Section 2.

As a test case, we examine the reaction of $^{137}Ce$ in collisions with impact parameters ranging from $b = 0$ fm to $b = 3$ fm, at a center-of-mass energy of 33 AMeV. For this proof of concept, 30 collisions were performed, each with random rotations of the

colliding nuclei; Figure 6 shows a sequence of snapshots of one of such collisions. The nuclei were prepared cold as explained in Section 3, and the initial separation of the colliding nuclei was 40 fm. The evolution of the collision was simulated using molecular dynamics integrated with the Leapfrog/Midpoint Euler method. The evolution time was 300 MeV/c, and the position and momenta of each nucleon was registered at intervals of 10 MeV/c. The identification of fragments was carried out by means of a minimum spanning tree algorithm, using a cut-off distance of 10 fm, eliminating loops between nucleons.

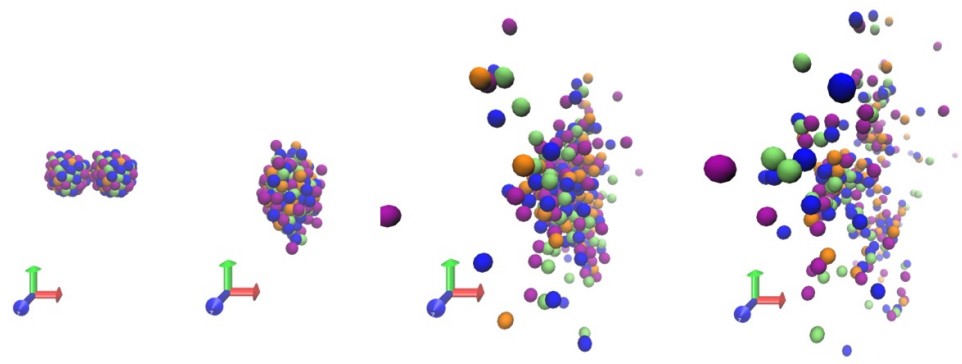

**Figure 6.** Four snapshots of the reaction $^{137}Ce + {}^{137}Ce$ at 33 AMeV and zero impact parameter.

The evolution of the reaction can be analyzed by tracking the average size of the fragments being produced in the reaction. Figure 7 shows the average (over 30 collisions) of the size of all fragments being produced, $\mu$, as a function of time. Clearly, at early times, say $t < 30$ MeV/c, the two $^{137}Ce$ nuclei merge to produce a single fragment of $\approx 2 \times 137$ with negligible dispersion $\sigma$. As evaporation and fragmentation set in, intermediate-mass fragments are produced and continue to evaporate.

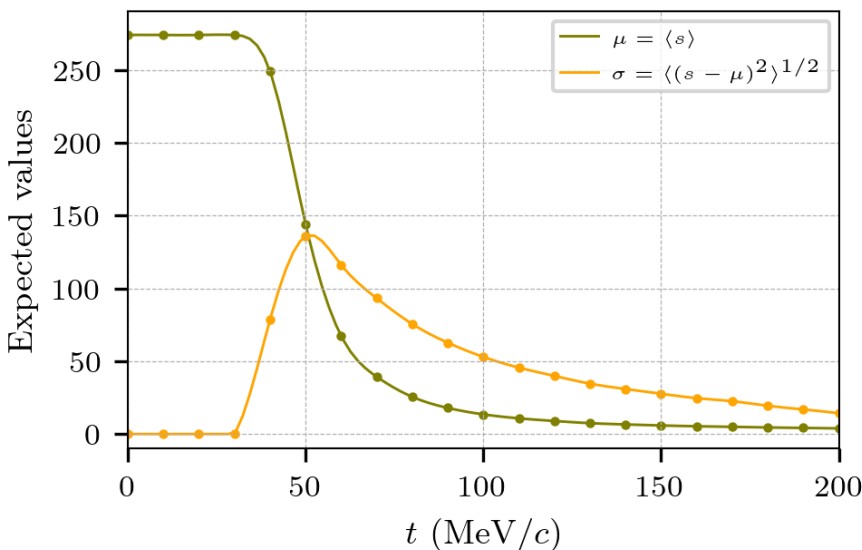

**Figure 7.** Evolution of the average size of fragments being produced, $\mu$, as a function of time, in 30 reactions of $^{137}Ce + {}^{137}Ce$ at 33 AMeV and $b = 0$. In addition, the dispersion $\sigma$ is shown.

The fragment multiplicity existing at different times during the collision is shown in Figure 8. The top panel corresponds to early times when the colliding nuclei merge together, the middle one is the mass distribution when the dispersion $\sigma$ reaches its maximum, and the bottom one is more similar to the asymptotic case without any large fragment left.

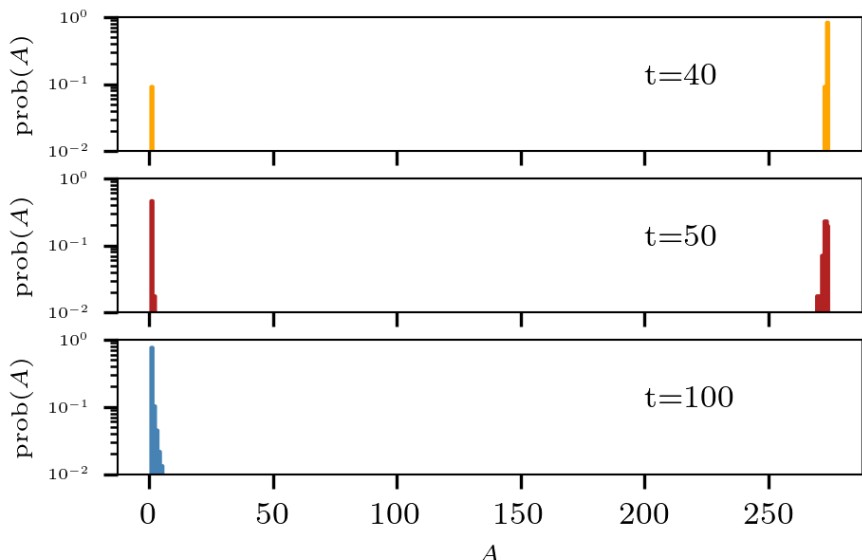

**Figure 8.** Fragment multiplicity at different times during the collision, obtained from 30 reactions of $^{137}Ce + {}^{137}Ce$ at 33 AMeV and $b = 0$.

Likewise, if we look at all the fragments produced during the entire evolution of the collisions, we find what is presented in Figure 9. The entire multiplicity obtained in the 30 collisions of $^{137}Ce + {}^{137}Ce$ at 33 AMeV, and $b = 0$ shows a clearly noticeable log-log relationship with the mass fragments. The particles produced with $A = 1$ until $A \approx 100$ are produced by evaporation, and the fragments larger than $A \approx 100$ were produced early in the reaction.

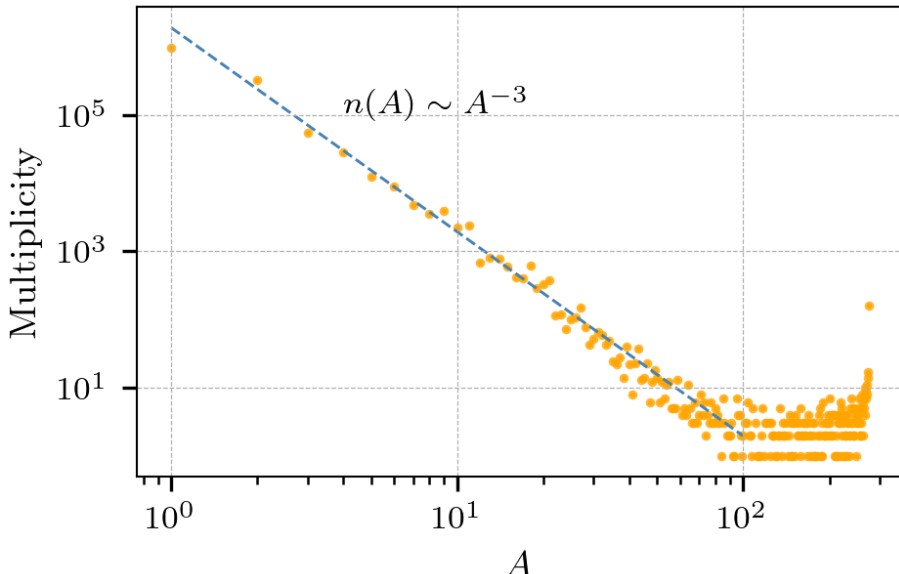

**Figure 9.** Added multiplicity of 30 collisions of $^{137}Ce + {}^{137}Ce$ at 33 AMeV and $b = 0$. Clearly noticeable is a power law relationship of the form $n = n_0 s^{-3}$ between the multiplicity $n$ and the mass fragments $s$.

Repeating the collisions for impact parameters $b = 0\ 1, 2$, and 3 fm, and for 50 collisions, we obtain similar multiplicities, as seen in Figure 10. The power law, however, is lost for intermediate mass fragments of $A > 20$. Indeed, the power law wanes as the impact parameter increases away from head-on collisions. This is due to the fact that, as $b$

increases, more nucleons become simple observers and do not participate in the sharing of momenta and energy; this effect produces an extreme "bump" for $b = 3$ fm at around $A$ 30.

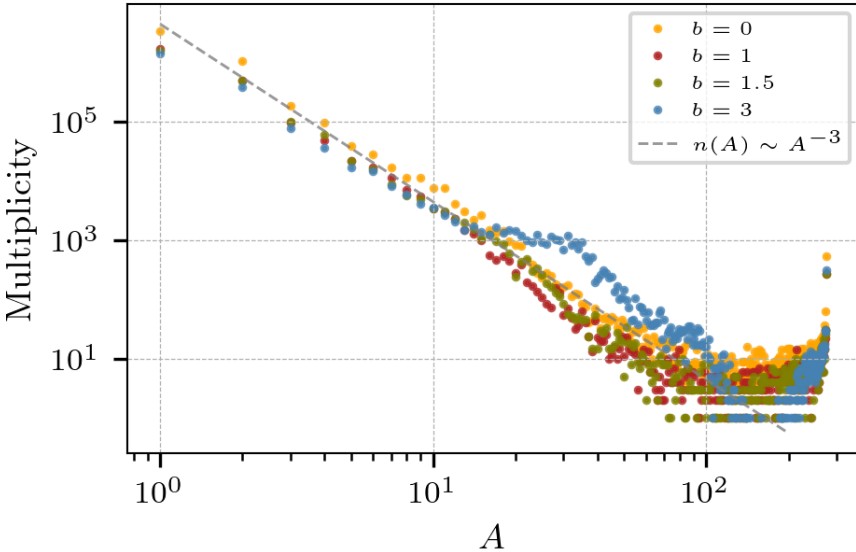

**Figure 10.** Added multiplicity of 30 collisions of $^{137}Ce + ^{137}Ce$ at 33 AMeV and various impact parameters. The power law relationship is lost for fragments of $A < 20$.

At a difference from the potential proposed in [28], our Pauli potential, Equation (1), contains only repulsive forces between equal nucleons, and thus de-enhances the formation of unphysical species such as di-neutrons, tri-neutrons, etc. To verify this, we examined the particle production formed during the evolution of 50 collisions of $^{137}Ce + ^{137}Ce$ at 33 AMeV and at $b = 0$ fm. Figure 11 shows the relative abundance of di-neutrons, tri-neutrons, etc. compared to the free neutrons. It is easy to see that the number of di-neutrons that appear during the collisions is two orders of magnitude less than the production of neutrons, and that higher multi-neutron compounds have even smaller probabilities of being produced; these exotic objects are energetically unstable and exist only for short times during the reaction.

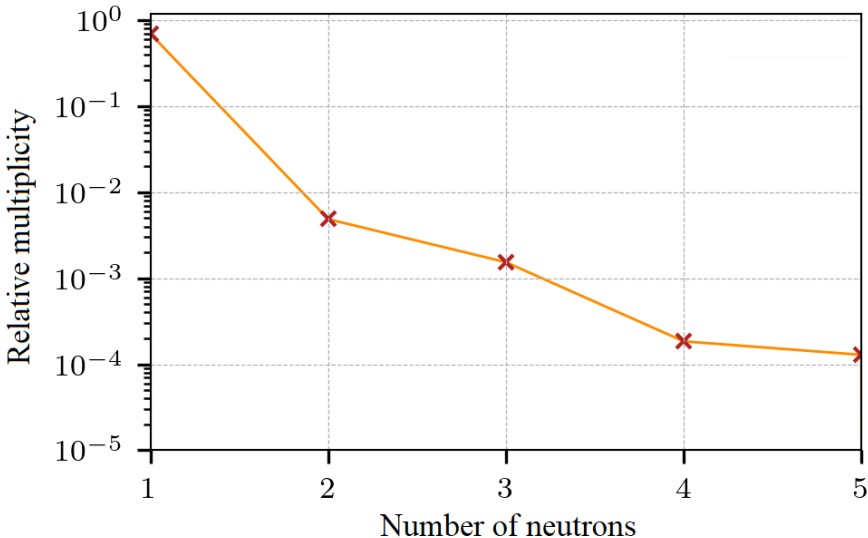

**Figure 11.** Relative production of pure neutron compounds formed during the evolution of 50 collisions of $^{137}Ce + ^{137}Ce$ at 33 AMeV and at $b = 0$ fm, normalized to the number of free neutrons.

## 6. Conclusions

In this work, we introduced a classical nucleon–nucleon potential that depends on the relative momenta of the colliding nucleons. The potential has the ability to mimic the Pauli exclusion principle by reducing interactions at low momentum transfer, preserving volume in phase space.

The potential was used to construct "nuclei" using a method of successive cooling and heating until reaching appropriate low energies. The parameters of the potential were adjusted to attain appropriate values of their binding energies and radii and, as displayed in Figures 1, 3, 5, and Table 1, the resulting values are quite acceptable. Similarly, the symmetry energy of the self-bound nuclei was calculated, yielding values in the range comparable to other estimations.

Besides the static properties, the nuclei were also used to study collisions. Resorting to CMD calculations with a Midpoint Euler method, a quasi simplectic evolution was simulated. Several relevant observables were studied, namely, the fragment mass distributions during the evolution, the time-evolution of the average size of fragments, and the fragment multiplicity at asymptotic times. Finally, in these preliminary results, we have found that production of neutron clusters is irrelevant.

In future studies, we will focus on collisions and direct comparison to experimental data.

**Author Contributions:** Conceptualization, C.O.D.; methodology, C.O.D.; software, C.O.D. and G.F; validation, C.O.D., G.F. and J.A.L.; formal analysis, C.O.D., G.F. and J.A.L.; investigation, C.O.D., G.F. and J.A.L.; resources, C.O.D. and G.F.; data curation, G.F.; writing—original draft preparation, J.A.L.; writing—review and editing, C.O.D., G.F. and J.A.L.; visualization, G.F.; supervision, C.O.D.; project administration, C.O.D.; funding acquisition, C.O.D. All authors have read and agreed to the published version of the manuscript.

**Funding:** C.O.D. received support from the Carrera de Investigador CONICET, by CONICET grants PIP0871, PIP 2015-2017 GI, founding D4247(12-22-2016), and Inter-American Development Bank (IDB), Grant No. PICT 1692.

**Data Availability Statement:** The data presented in this study (i.e., data from simulations) are available on request from the corresponding author.

**Conflicts of Interest:** The authors declare no conflict of interest.

## Appendix A

Equation (4) in Section 3 involves an explicit dependence on two variables, namely, A and Z, which introduces discontinuities for different isotopes of the same element; such fluctuations can be seen more noticeable around $A \approx 70$ in Figure 3. Because of this double A–Z dependence, to determine a fit to the simulated data, the following procedure was used:

1. The chosen model was $y = C_v x_0 - C_s x_1 - C_c x_2 - C_{sym} x_3$, where $x_0 = 1$ $x_1 = A^{-1/3}$, $x_2 = Z.A^{-4/3}$, and $x_3 = (A - 2Z)^2 A^{-2}$. This is a linear model with the parameters $C_v$, $C_s$, $C_c$, and $C_{sym}$ to be estimated. We intentionally avoided nonlinear models since these may lead to sub-optimal estimates. Notice, however, that the estimation is conditioned to the chosen isotopes ($A$, $Z$ values);

2. The estimation was carried out by means of a least-trimmed-square estimator, which corresponds to the class of robust estimators, and is known to be more insensitive to outliers;

3. The fitting shown in Figure 3 corresponds to the evaluation of the model at the $\{x_0, x_1, x_2, x_3\}$ data set. The values in between correspond to splines of order 2.

The resulting values of $C_v$, $C_s$, $C_c$, and $C_{sym}$ are listed in Table 2.

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
