# Peer review of "Pauli Exclusion Classical Potential for Intermediate-Energy Heavy-Ion Collisions"

_universe, doi:10.3390/universe9030119_

Round 1

Reviewer 1 Report

In this article the authors modelize the nucleon interaction using an essentially classical model through a potential which retains the main physical aspects of that interaction. They use then this potential to study the binding energy, radius as well as the main results of collisions between nuclei.

In my opinion the paper is well and clearly written. The proposed model fits quite accurately the experimental data. Taking into account the simplicity of the model, using classical potentials, the model illustrates remarkably well the different interactions that are relevant for the phenomena considered.

I recommend the publication of the article in its present form.

Author Response

Notes in pdf.

Reviewer 2 Report

I have found this paper very interesting, as it presents a semi-classical approach to the study of static and dynamic properties of nuclear systems. The extension of the model developed, validated on static properties, to collisions looks very promising as a complement to the quantum-mechanical approach.

The Authors propose a development of their previous works including the new ingredients to account for Pauli exclusion, and considering the dependence of the nuclear matter behaviour on small distance and momenta.

The ingredients of the model, after a very detailed and comprehensive introduction and discussion about the matching of semi-classical approach with typical quantum phenomena, are clearly introduced and commented. Figures chosen are a right complement to the text, helping very much the reader in catching the procedures' goals and their validation.

The whole discussion is properly referenced. There is a certain level of self-citations (<~30%), though they are basically justified by the apparent historical specialization of the Authors on these subject.

Refs. 18 and 33 seems the same. Ref. 19 is empty. Please revise.

The Conclusions appear to be somewhat cut off, and in my opinion should deserve some outlook not only on the further development of the same calculation to other collisions, but also on some hints of possible experimental or phenomenological verification. This is especially apparent if compared to the very detailed introduction of the article.

The English language appears to be generally good and plain, except for a very few typos (see the attachment, please revise).

In summary, I think this article eligible for publication, after minor revision and check Authors are invited to consider in the attachment.

Author Response

Notes on pdf.

Reviewer 3 Report

The manuscript is a continuation Ref. 47 from the same group and presents a refined classical nucleon nucleon repuslive potential within the classical dynamics approach  to hopefully mimic Pauli exclusion principle. The potential works surprising well in reproducing the binding energies, radii as well as the symmetry energy coefficients, though there is still a noticeable in the intermediate mass region. The manuscript is very well written and informative. The results are quite encouraging. Therefore I support its prompt publications.

A few minor comments for the authors to consider:

1) I think it is still necessary to specify in the title and abstract that the potential is within the classical dynamics.

2) It should help the readers to specify what are BUU and QMD.

3)Eq. 1,  with q0=6 it means that the potential is nearly a constant within the nucleus and extends quite further out. Can the authors comment on how sensitive their results are to q0 and the cutoff (if one reintroduces it). And to how much extent the repulsive potential simulates the Pauli principle or just a correction to the nn/pp potential.

4) Fig. 4, I wonder how reliable is the fitting to the simulation since naively looking there is quite noticeable fluctation around A=70.

5) Fig5, There are experimental data for Cs isotopes for the range of x shown. I recommend the authors to add the experimental data and more calculations if possible. Otherwise the extrapolation may look quite risky as it is now.

6) Conclusion: a classical nucleon-nucleon-> a classical nucleon-nucleon repulsive potential

Author Response

Notes on pdf.

Round 2

Reviewer 2 Report

I thank the Authors for having widely addressed the reviewers' comments and suggestions, thus improving the overall quality of the work. 

Reviewer 3 Report

I recommend the prompt publication of the revised version